# A New Three-Way Incremental Naive Bayes Classifier

Zhiwei Yang [1], Jing Ren [1,2], Zichi Zhang [3], Yuqing Sun [4], Chunying Zhang [1,5,6,7], Mengyao Wang [1] and Liya Wang [1,5,6,7,*]

1. College of Science, North China University of Science and Technology, Tangshan 063210, China; yangzhiwei@stu.ncst.edu.cn (Z.Y.); renjing8778@163.com (J.R.); zchunying@ncst.edu.cn (C.Z.)
2. College of Computer Science and Mathematics, Anyang University, Anyang 455000, China
3. College of Electrical Engineering, North China University of Science and Technology, Tangshan 063210, China; zhangzichi614@stu.ncst.edu.cn
4. College of Economics, North China University of Science and Technology, Tangshan 063210, China; sunyuqing@stu.ncst.edu.cn
5. Hebei Key Laboratory of Data Science and Application, North China University of Science and Technology, Tangshan 063210, China
6. The Key Laboratory of Engineering Computing in Tangshan City, North China University of Science and Technology, Tangshan 063210, China
7. Tangshan Intelligent Industry and Image Processing Technology Innovation Center, North China University of Science and Technology, Tangshan 063210, China
* Correspondence: wangliya@ncst.edu.cn

**Abstract:** Aiming at the problems of the dynamic increase in data in real life and that the naive Bayes (NB) classifier only accepts or rejects the sample processing results, resulting in a high error rate when dealing with uncertain data, this paper combines three-way decision and incremental learning, and a new three-way incremental naive Bayes classifier (3WD-INB) is proposed. First, the NB classifier is established, and the distribution fitting is carried out according to the minimum residual sum of squares (RSS) for continuous data, so that 3WD-INB can process both discrete data and continuous data, then carry out an incremental learning operation, select the samples with higher data quality according to the confidence of the samples in the incremental training set for incremental learning, solve the problem of data dynamics and filter the poor samples. Then we construct the 3WD-INB classifier and determine the classification rules of the positive, negative and boundary domains of the 3WD-INB classifier, so that the three-way classification of samples can be realized and better decisions can be made when dealing with uncertain data. Finally, five discrete data and five continuous data are selected for comparative experimental analysis with traditional classification methods. The results show that 3WD-INB has high accuracy and recall rate on different types of datasets, and the classification performance is also relatively stable.

**Keywords:** naive Bayes; three-way decision; incremental learning; 3WD-INB; distribution fitting

## 1. Introduction

The classification problem is a foundation in the field of data mining, but it is also a very important means. Common traditional classifiers include naive Bayes (NB), random forest (RF), support vector Mac (SVM), K-nearest neighbors (KNN), multilayer perceptron classifier (MLP), etc. In recent years, many scholars have made great progress in the research of new classifiers and created many new classifiers [1–4].

The naive Bayes classifier (NB) was first proposed by Duda and Hart in 1973. Its core idea is to calculate the probability that the sample belongs to each category given the characteristic value of the sample and assign it to the category with the highest probability. This algorithm does not require a large amount of training data and has good interpretability, so it has attracted the attention and use of more and more researchers. In summary, the naive Bayes classifier has the following advantages:

1. It performs well on small-scale data and can not only handle binary classification tasks but also multi-classification tasks.
2. The algorithm is simple to establish and less sensitive to missing datasets.
3. It has high speed for large-scale training and query and is suitable for large-scale datasets.

Therefore, naive Bayes is widely used and has achieved good results in text classification, spam email filtering, medical diagnosis, and other fields. To eliminate the zero probability and over-fitting problems in naive Bayes classification, Xu et al. [5] designed two smoothing strategies, M-estimation and Laplace estimation, which effectively improved the classification performance. Li et al. [6] used Pearson and Kendall coefficients to screen out new attribute sets based on principal component analysis to make them meet the conditional independence assumption as much as possible and constructed NB-IPCA classifiers to improve the classification accuracy. Farid et al. [7] proposed a hybrid decision tree and a hybrid naive Bayes classification algorithm and solved the multi-classification problem. For text classification problems, Zhang et al. [8] created a two-layer Bayes model: random forest naive Bayes (RFNB); the first layer is a random forest model, and the second layer is a Bernoulli naive Bayes model. Gama et al. [9] proposed an adaptive Bayes model, which is an incremental learning algorithm that can work online, and has improved performance compared with nonadaptive algorithms. Li et al. [10] used the weighted K-nearest neighbor algorithm to calculate the membership degree of unlabeled samples and improved the structure of the naive Bayes classifier through the membership degree to optimize its classification effect. Qiu et al. [11] combined the particle swarm optimization algorithm with naive Bayes, which effectively reduced redundant attributes and improved the classification ability. Ramoni et al. [12] constructed a robust Bayes classifier (RBC) for datasets with missing values, which can handle incomplete databases without assuming missing data patterns. Zhang et al. [13] proposed an attribute enhancement and weighted naive Bayes algorithm, which can find potential attributes beyond the original attribute space and is used to solve the attribute conditional independence assumption, and experiments have proved that the algorithm has achieved good results. Kaur et al. [14] used the weighted information gain method to reassign the features of the misclassified classifications and combined it with the polynomial naive Bayes classification algorithm to provide a better classification. For naive Bayes to be applied to continuous data, Fisher [15] assumes that the probability distribution for each classification is Gaussian (also known as normal distribution), treats multiple measurements as random variables and estimates the probability using a Gaussian function. Fisher [16] also proposed the method of discretizing continuous data for the first time. Since then, this method has been widely used in various fields, including machine learning, data mining, statistics, and so on, including the naive Bayes classifier. Fayyad et al. [17] improved the naive Bayes under interval discretization and used the basic principles of information theory to guide the operation of the multi-interval discretization process, and the results showed that the new method has significantly improved the classification accuracy. However, the traditional Bayes classifier still belongs to the two-way decision model; that is, there are two kinds of processing for the classification results of samples: accepting or rejecting. When dealing with uncertainties, the inability to make accurate decisions on samples will lead to poor classification performance. The three-way decision has the characteristics that conform to human thinking and cognition and can better handle the uncertainties in the actual decision-making process. Therefore, some scholars have improved the naive Bayes algorithm with the three-way decision. Zhang et al. [18] constructed a new three-way extended TAN Bayes classifier combining the three-way decision thinking and considering the attribute condition independence, which effectively improved the classification performance. Zhou et al. [19] combine three-way decision with the naive Bayes classifier and use it to classify junk email. In addition to classifying normal email and junk email, users are allowed to further check for uncertain email, which has been experimentally proven to reduce the rate of misclassification. Later, Zhang et al. [20] integrated naive Bayes, three-way decision and collaborative filtering algorithm, and proposed a three-way decision naive Bayes collaborative filtering recommendation (3NBCFR) model,



which was used for a movie recommendation, effectively reducing the cost of recommendation and improving the quality of the recommendation. However, the above improvements to the Bayes classifier also have the following practical problems:

(1) Datasets in the real world are generally generated dynamically, and the amount of data is constantly changing. It is difficult to obtain credible posterior probabilities based on limited training sets, and it is time consuming to reuse new datasets for training.

(2) Most Bayes classifiers are generally applied to discrete data, and the scope of application of the model is small. The traditional improvement method is to discretize continuous data or use the Gaussian function, but the former is difficult to set the discrete interval, and the latter has high requirements on the distribution of datasets, neither of which can solve the classification problem of continuous data well.

To this end, this paper combines incremental learning, three-way decision and naive Bayes classifier and proposes a new three-way decision incremental naive Bayes (3WD-INB) classifier. The contributions of this paper are as follows:

(1) Combining three-way decision ideas with the traditional naive Bayes classifier, which makes the decision-making mode of the classifier more in line with the human thinking process and improves the classification ability of uncertain data.

(2) The incremental learning method solves the problem of data dynamics, and at the same time, it can filter the poor-quality data samples and optimize the training data in the incremental learning stage.

(3) For continuous data, the distribution of data is fitted according to the sum of squares of minimum residuals (RSS), and the posterior probability is estimated by the distribution function, so that 3WD-INB can be applied not only to discrete data but also to continuous data, which enhances the applicability of the classification models.

(4) Compared with the traditional Bayes model and other traditional classification models, 3WD-INB effectively improves the classification performance. Relative to the NB classifier, *F*1 is increased from 0.6364 to 0.9167 in discrete data and *precision* from 0.7778 to 1.0000 in discrete data. Relative to the G-NB classifier, with continuous data, *F*1 increased from 0.8036 to 0.9967 and *precision* from 0.5285 to 0.8850. The average *F*1 of 3WD-INB under discrete and continuous data are 0.9501 and 0.9081, respectively, and the average *precision* is 0.9648 and 0.9289, respectively.

The structure of this paper is as follows. In the second section, the current naive Bayes classifier and the relevant content of the three-way decision are introduced, and the basic theory is explained to provide a theoretical basis for subsequent models and algorithms. In the third section, the distribution fitting process, incremental learning process, classification rule derivation process and overall algorithm steps of the 3WD-INB classifier are explained in detail. In the fourth section, the parameter change analysis experiment is carried out on 3WD-INB, and different types of datasets are selected for comparative experiments with traditional algorithms. In the last section, the full text is summarized, and future work is proposed.

## 2. Related Work

### 2.1. Naive Bayes Classifier

The naive Bayes theory is based on the Bayes theorem and has a sufficient basis in probability theory. It first classifies by constructing a Bayes classifier structure and then by calculating the posterior probability of each object.

Given a training set with a sample size of $N$: $U = \{x_1, x_2, \cdots, x_N\}$, the training set contains $n$ attributes: $A = \{a_1, a_2, \cdots, a_n\}$; the category of the data label is $k$: $C_1, C_2, \cdots, C_k$. We express the training sample $x_h$ as a $n$-dimensional feature vector $x_h = \left\{ v_{1,}^{(h)} v_2^{(h)}, \cdots, v_n^{(h)} \right\}$; $v_i^{(h)}$ represents the value of sample $x_h$ in attribute $a_i$. Then according to Bayes theorem, the posterior probability $P(C_c|x_h)$ can be obtained, as shown in Equation (1).

$$P(C_c|x_h) = \frac{P(C_c)P(x_h|C_c)}{P(x_h)} \tag{1}$$

where $P(C_c)$ is the prior probability, $P(x_h|C_c)$ is the conditional probability, and $P(x_h)$ is a constant.

For the NB classifier, its probability estimation expression is shown in Equation (2).

$$P(C_c|x_h) \propto P(C_c)\prod_{i=1}^{n} P\left(v_i^{(h)}\middle|C_c\right) \tag{2}$$

where $n$ represents the number of attributes, and $v_i^{(h)}$ represents the value of $x_h$ on the $i$-th attribute $a_i$.

In the classification process of the naive Bayes classifier, the calculation formulas of $P(C_c)$ and $P\left(v_i^{(h)}\middle|C_c\right)$ are shown in Equations (3) and (4).

$$P(C_c) = \frac{|C_c|}{|U|}, c = 1, 2, 3, \cdots, k \tag{3}$$

$$P\left(v_i^{(h)}\middle|C_c\right) = \frac{\left|m\left(a_i, v_i^{(h)}\right) \cap C_c\right|}{|C_c|}, c = 1, 2, 3, \cdots, k \tag{4}$$

where $|U|$ is the total number of training samples, $|C_c|$ is the number of samples of category $C_c$ in the training samples, and $m\left(a_i, v_i^{(h)}\right) \cap C_c$ represents the set of objects in $C_c$ that take the value of $v_i^{(h)}$ on the $i$-th attribute.

At the same time, in the Bayes classifier, to avoid the value not appearing in the training sample in the test sample, resulting in $\left|m\left(a_i, v_i^{(h)}\right) \cap C_c\right| = 0$, using Laplace smoothing operation, the calculation formula of the sum after smoothing is shown in Equations (5) and (6).

$$P(C_c) = \frac{|C_c| + 1}{|U| + k}, c = 1, 2, 3, \cdots, k \tag{5}$$

$$P\left(v_i^{(h)}\middle|C_c\right) = \frac{\left|m\left(a_i, v_i^{(h)}\right) \cap C_c\right| + 1}{|C_c| + |a_i|}, c = 1, 2, 3, \cdots, k \tag{6}$$

Finally, use the smoothed $P(C_c)$ and $P\left(v_i^{(h)}\middle|C_c\right)$ to classify the samples and obtain the classification label $H(x_h)$; the formula is Equation (7).

$$H(x_h) = \arg\max P(C_c)\prod_{i=1}^{n} P\left(v_i^{(h)}\middle|C_c\right) \tag{7}$$

When the data is continuous, the above method is no longer applicable to the calculation of $P\left(v_i^{(h)}\middle|C_c\right)$, and the usual solutions are as follows:

2.1.1. Interval Continuous Data (D-NB)

Commonly used continuous data interval methods include the following:

(1) Equal width interval method: divide the data value range into several intervals equally, and the width of each interval is equal.
(2) Equal frequency interval method: divide the data range into several intervals, and each interval contains an equal amount of data.
(3) Clustering-based method: use a clustering algorithm to cluster continuous data into several groups, and the data in each group are regarded as the same discrete value.
(4) Method based on information entropy: use information entropy to measure the information gain of each division point and select the division point with the largest information gain as the dividing point of discretization.

These methods have their own advantages and disadvantages, and the specific choice should be considered according to the actual situation. Here, we mainly introduce the equal width interval method, which is also the simplest and most commonly used method. The general idea is to divide the continuous data feature $v_i^{(h)}$ into $K$ intervals $D = \{d_1, d_2, \cdots, d_K\}$. For example, suppose the original data are $v_{\text{original}} = \{1, 2.5, 3, 4.75, 5, 6, 7, 8, 9, 9.9\}$. The specified intervals are as follows: $d_1 = [0, 2]$, $d_2 = (2, 4]$, $d_3 = (4, 6]$, $d_4 = (6, 8]$, $d_5 = (8, 10]$; then, the internalized data are $v_{new} = \{1, 1, 2, 2, 3, 3, 4, 4, 5, 5\}$. After such processing, the continuous data are successfully transformed into discrete data, and the calculation problem of $P\left(v_i^{(h)} \middle| C_c\right)$ is also solved. However, this method has high requirements for the number of intervals and the division method of the intervals, and it is difficult to determine the optimal value of the two in the actual operation process.

2.1.2. Gaussian Naive Bayes (G-NB)

Assuming that the data of each dimension satisfy the normal distribution, that is, $P\left(v_i^{(h)} \middle| C_c\right) \sim N(\mu_i, \sigma_i)$, where $\mu_i$ and $\sigma_i$ are the mean and variance of $v_i^{(h)}$, respectively, then:

$$P\left(v_i^{(h)} \middle| C_c\right) = \frac{1}{\sqrt{2\pi}\sigma_i} e^{-\frac{\left(v_i^{(h)} - \mu_i\right)^2}{2\sigma_i^2}} \tag{8}$$

Therefore, according to the naive Bayes classifier (NB), the final classifier formula of G-NB is shown in Equation (9).

$$H(x_h) = \arg\max P(C_c) \prod_{i=1}^{n} \frac{1}{\sqrt{2\pi}\sigma_i} e^{-\frac{\left(v_i^{(h)} - \mu_i\right)^2}{2\sigma_i^2}} \tag{9}$$

G-NB solves the problem of continuous data processing, but the assumption that all dimensional data satisfy the normal distribution has high requirements for the dataset, and the datasets in the real world are often distributed in multiple ways, so the application effect of G-NB is not good, unstable. Therefore, this paper proposes a new method to deal with continuous data to improve the robustness of the classifier.

*2.2. Three-Way Decision*

Three-way decision is a decision-making method summarized by Yao [21] in the research process of rough set theory. The general idea is to divide the universe into three (positive domain, negative domain and boundary domain) and adopt different decision-making methods for different domains, as shown in Figure 1, which is more in line with human thought and cognition. In recent years, domestic and foreign scholars have proposed a series of three-way decision models, which have been widely used in medical diagnosis [22], garbage mailbox prediction [23] and other disciplines and fields. Liu et al. [24] systematically introduced the theory, method and application of the integration of three-way decision and rough set theory from the perspective of three-way decision. Liu et al. [25] also proposed broad and narrow theoretical models of three-way decision from the macro and micro perspectives. The broad three-way decision focuses on the interpretation of the connotation and extension of the concept of three-way decision, and the narrow three-way decision focuses on the semantic interpretation of three-way decision in practical decision-making problems. Yao et al. [26] explained and analyzed the basic concepts and theories in formal concept analysis, rough set and granular computing in detail and pointed out the relationship between them. Liang et al. [27–30] substituted fuzzy concepts such as interval value, triangular fuzzy number and intuitionistic fuzzy set for the precise conditional probability function in the three-way decision, making the three-way decision model more widely used. Yang et al. [31] proposed a new fuzzy rough set model based on three-way decision with optimal similarity, which makes the model

more robust to noise and beneficial to the application of the fuzzy information systems. In Long et al. [32], to further introduce the fuzzy set theory into the three-way decision concept analysis, the attribute-derived fuzzy three-way decision and object-derived fuzzy three-way decision are studied under the background of the fuzzy form. The existing classical three-way decision is extended to the fuzzy three-way decision, which is important to improve the three-way decision theory. Xue et al. [33] proposed a three-way decision model based on probability graphs by constructing the Bayes network to calculate the conditional probability distribution function. Jia et al. [34] proposed a feature fusion method based on the three-way decision model, which combines a single feature extraction method and multiple feature extraction methods to maximize the use of different feature information and improve the performance of Chinese satire detection. Dai et al. [35] proposed a new three-way decision model. Unlike the traditional three-way decision model, this model uses intuitive fuzzy sets to describe the attributes of decision objects and preferences of decision-makers and concept lattice theory to represent the relationship between attributes to better deal with uncertainty and fuzziness. Many scholars have also applied the three-way decision to the classification model. Li et al. [36] transformed the problem of software defect prediction into three kinds of decision-making using three-way decision methods: identifying defective software, identifying defective software and identifying uncertain software, which have better accuracy and reliability. Chen et al. [37] constructed an emotional analysis model based on three-way decision, which solves the problem of "unknown" emotions by introducing an intermediate category and also uses the flexibility of three-way decision to classify different emotional intensities, which has higher accuracy in emotional classification. Chu et al. [38] proposed a three-component clustering method based on neighborhood rough set to study the classification of gout patients. This method can deal with uncertain data, which is very useful for applications such as medical diagnosis. Wang et al. [39] proposed an adaptive weighted three-way decision oversampling method to solve the problem of unbalanced data classification. This method uses the idea of three-way decision, combines the oversampling technology with clustering technology, and can more effectively identify a few classes of samples and improve the classification accuracy when dealing with unbalanced data. Remesh et al. [40] proposed a three-way decision technology based on variance criteria to detect COVID-19 patients. Finally, patients can be divided into three categories: confirmed, suspected and non-COVID-19. Confirmed samples can be treated, and suspected samples can be further detected, which has potential application value in early diagnosis and screening of COVID-19.

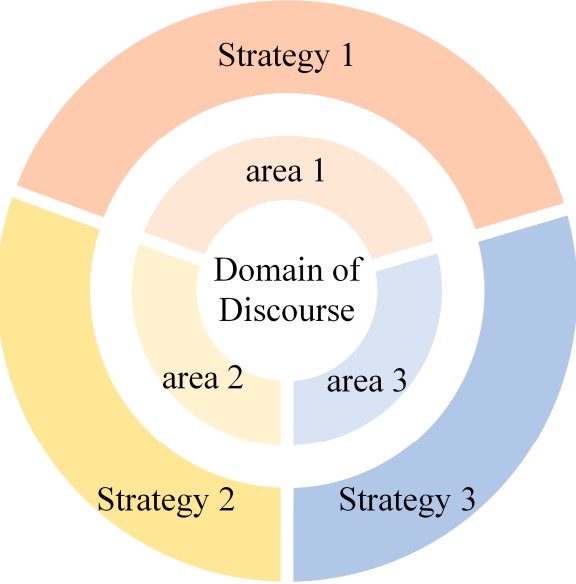

**Figure 1.** The idea of three-way decision.

At present, the three-way decision has become the focus of many scholars. They have been widely used in information system analysis, machine learning models and artificial intelligence decision-making and have achieved good results in theory. Nevertheless, the three-way decision is largely limited by data quality and quantity. When processing data, the three-way decision method needs to fully consider the limitation of data quality and quantity. If the data quality is poor or the amount of data is small, the accuracy and reliability of the three-way decision may be affected.

### 2.3. The Basic Theory of Three-Way Decision

Given an information system, $S = (U, A \bigcup D, \{V_a | a \in A\}, \{I_a | a \in A\})$, for $C \subseteq U$, set $D = \{C, \overline{C}\}$ to represent two states (respectively indicate whether it belongs to set $C$, and $C$ and $\overline{C}$ are complementary), and set $AC = \{a_P, a_N, a_B\}$ represents three decision-making actions (respectively representing acceptance, rejection and delay) and decision-making action. The cost function is shown in Table 1.

**Table 1.** Cost Function.

| Decision Making | $C$ (Positive Example) | $\overline{C}$ (Negative Example) |
|---|---|---|
| $a_P$ | $\lambda_{PP}$ | $\lambda_{PN}$ |
| $a_N$ | $\lambda_{NP}$ | $\lambda_{NN}$ |
| $a_B$ | $\lambda_{BP}$ | $\lambda_{BN}$ |

The expected costs $R(a_P | x_h)$, $R(a_N | x_h)$ and $R(a_B | x_h)$ of the three decisions $a_P, a_N$ and $a_B$ are shown in Equation (10).

$$R(a_P | x_h) = \lambda_{PP} P(C | x_h) + \lambda_{PN} P(\overline{C} | x_h)$$
$$R(a_N | x_h) = \lambda_{NP} P(C | x_h) + \lambda_{NN} P(\overline{C} | x_h)$$
$$R(a_B | x_h) = \lambda_{BP} P(C | x_h) + \lambda_{BN} P(\overline{C} | x_h) \tag{10}$$

where $P(C | x_h)$ represents the conditional probability that $x_h$ belongs to the set $C$, and $P(\overline{C} | x_h)$ represents the conditional probability that $x_h$ belongs to the set $\overline{C}$.

According to the basic theory of three-way decision, use the expected cost $R(a_P | x_h)$, $R(a_N | x_h)$ and $R(a_B | x_h)$ to make an action decision on $x_h$, and the minimum cost decision [41] rule is as follows:

(P) If $R(a_P | x_h) \leq R(a_N | x_h)$ and $R(a_P | x_h) \leq R(a_B | x_h)$, then for $x_h$, there is $x_h \in POS_{(\alpha, \beta)}(C_c)$: accept the decision;

(N) If $R(a_N | x_h) \leq R(a_P | x_h)$ and $R(a_N | x_h) \leq R(a_B | x_h)$, then for $x_h$, there is $x_h \in NEG_{(\alpha, \beta)}(C_c)$: reject the decision;

(B) If $R(a_B | x_h) \leq R(a_P | x_h)$ and $R(a_B | x_h) \leq R(a_N | x_h)$, then for $x_h$, there is $x_h \in BND_{(\alpha, \beta)}(C_c)$: delay the decision.

In addition, because $P(C | x_h) + P(\overline{C} | x_h) = 1$, $\lambda_{PP} \leq \lambda_{BP} \leq \lambda_{NP}$, $\lambda_{PN} \leq \lambda_{BN} \leq \lambda_{NN}$, the rules can be further simplified:

(P) If $P(C | x_h) \geq a$ and $P(C | x_h) \geq \gamma$, then for $x_h$, there is $x_h \in POS_{(\alpha, \beta)}(C_c)$: accept the decision;

(N) If $P(C | x_h) \leq \beta$ and $P(C | x_h) \leq \gamma$, then for $x_h$, there is $x_h \in NEG_{(\alpha, \beta)}(C_c)$: reject the decision;

(B) If $P(C | x_h) \leq a$ and $P(C | x_h) \geq \beta$, then for $x_h$, there is $x_h \in BND_{(\alpha, \beta)}(C_c)$: delay the decision.

In the rules:

$$a = \frac{\lambda_{PN} - \lambda_{BN}}{(\lambda_{PN} - \lambda_{BN}) + (\lambda_{BP} - \lambda_{PP})}$$

$$\beta = \frac{\lambda_{BN} - \lambda_{NN}}{(\lambda_{BN} - \lambda_{NN}) + (\lambda_{NP} - \lambda_{BP})}$$

$$\gamma = \frac{\lambda_{PN} - \lambda_{NN}}{(\lambda_{PN} - \lambda_{NN}) + (\lambda_{NP} - \lambda_{PP})} \tag{11}$$

If it is further assumed that the cost function satisfies Equation (12), it can be proved that $a > \gamma > \beta$.

$$\frac{\lambda_{NP} - \lambda_{BP}}{\lambda_{BN} - \lambda_{NN}} > \frac{\lambda_{BP} - \lambda_{PP}}{\lambda_{PN} - \lambda_{BN}} \tag{12}$$

Continuing to simplify, the final three-way decision minimum cost decision rule is:
(P) If $P(C|x_h) \geq a$, then $x_h \in POS_{(\alpha,\beta)}(C_c)$: accept the decision;
(N) If $P(C|x_h) \leq \beta$, then $x_h \in NEG_{(\alpha,\beta)}(C_c)$: reject the decision;
(B) If $\beta < P(C|x_h) < a$, then $x_h \in BND_{(\alpha,\beta)}(C_c)$: delay the decision.

## 3. Three-Way Decision Incremental Naive Bayes Classifier

In the real world, the data acquisition process is often acquired dynamically. For the classification model, it will consume a lot of time to reuse new data for training. Considering that the three decision-making is more in line with the human thinking process, the incremental learning and the three-way decision are combined with the NB classifier to build a three-way decision incremental naive Bayes classifier (3WD-INB). In addition, considering that most of the naive Bayes classifiers are used for discrete data in general and that the classification problem of continuous data is also very common in the real world, the continuous data are distributed using the minimum residual sum of squares (RSS). The fitting process makes 3WD-INB also able to deal with the classification problem of continuous data.

### 3.1. Improvements for Continuous Data

For discrete data, $P\left(v_i^{(h)}\middle|C_c\right)$ can be calculated directly according to Equation (6), but for continuous data, $P\left(v_i^{(h)}\middle|C_c\right)$ cannot be calculated in an original way. For this purpose, a distribution function is fitted to the distribution of $v_i^{(h)}$ according to the residual sum of squares, then the distribution function is used to find the $P\left(v_i^{(h)}\middle|C_c\right)$ of the continuous data.

We refer to the Distfit official [42], and we selected 10 common distributions to apply to our model and used *RSS* to fit the distribution. The 10 distributions are shown in Table 2.

**Table 2.** Selected distribution type.

| Serial Number | 1 | 2 | 3 | 4 | 5 | 6 | 7 | 8 | 9 | 10 |
|---|---|---|---|---|---|---|---|---|---|---|
| Distribution | norm | expon | pareto | dweibull | t | genextreme | gamma | lognorm | beta | uniform |

*RSS* describes the predicted deviation from the actual empirical value of the data. It is a measure of the difference between the data and the estimated model. A small *RSS* indicates a good fit of the model to the data. The calculation formula of *RSS* is shown in Equation (13).

$$RSS = \sum_{i=1}^{n}(y_i - f(x_i))^2 \tag{13}$$

where $y_i$ is the $i$ value of the variable to be predicted, $x_i$ is the $i$ value of the explanatory variable, and $f(x_i)$ is the predicted value of $y_i$.

During the fitting process, calculate the value of *RSS* under 10 distributions: $\{rss_1, rss_2, \cdots, rss_{10}\}$. The distribution with the smallest *RSS* is the fitting optimal distribution; that is, the optimal distribution is arg min $\{rss_1, rss_2, \cdots, rss_{10}\}$.

Given a training set with $N$ samples, $U = \{x_1, x_2, \cdots, x_N\}$, the training set contains $n$ attributes, $A = \{a_1, a_2, \cdots, a_n\}$, and the category of data labels has $k$ categories, $C_1, C_2, \cdots, C_k$. Represent the training sample $x_h$ as an $n-$dimensional feature vector $x_h = \left\{v_{1,}^{(h)} v_2^{(h)}, \cdots, v_n^{(h)}\right\}$; $v_i^{(h)}$ represents the value of the sample $x_h$ in attribute $a_i$.

The specific operation of the fitting is as follows (Algorithm 1):

---

**Algorithm 1**: My Distfit

---

**Input**: Training set: $U = \{x_1, x_2, \cdots, x_N\}$

**Output**: Fitted distribution function: $p_i\left(v_i^{(h)}|C_c\right), p_i\left(v_i^{(h)}|\overline{C}_c\right)$

1. **For** $v_i^{(h)} \in x_h$ **do**
2. **For** $i = 1, 2, \cdots, N$ **do**
3. Calculate the set of $RSS_{C_c}$ for 10 distributions: $\left\{rss_{C_c 1}, rss_{C_c 2}, \cdots, rss_{C_c 10}\right\}$
4. $p_i\left(v_i^{(h)}|C_c\right) = $ arg min $\left\{rss_{C_c 1}, rss_{C_c 2}, \cdots, rss_{C_c 10}\right\}$
5. Calculate the set of $RSS_{\overline{C}_c}$ for 10 distributions: $\left\{rss_{\overline{C}_c 1}, rss_{\overline{C}_c 2}, \cdots, rss_{\overline{C}_c 10}\right\}$
6. $p_i\left(v_i^{(h)}|C_c\right) = $ arg min $\left\{rss_{\overline{C}_c 1}, rss_{\overline{C}_c 2}, \cdots, rss_{\overline{C}_c 10}\right\}$
7. **End for**
8. **End for**
9. **Return** $p_i\left(v_i^{(h)}|C_c\right), p_i\left(v_i^{(h)}|\overline{C}_c\right)$

---

The optimal distribution is obtained by fitting with Algorithm 1: $p_i\left(v_i^{(h)}|C_c\right), p_i\left(v_i^{(h)}|\overline{C}_c\right)$, approximately estimating $P\left(v_i^{(h)}|C_c\right)$ and $P\left(v_i^{(h)}|\overline{C}_c\right)$ for continuous data.

### 3.2. Incremental Features

This part is the process of building the INB classifier. Because of the dynamic nature of the dataset, this paper uses the incremental feature of naive Bayes and adopts incremental learning. In addition, incremental learning can filter samples with high data quality, which can improve the performance of the model to a certain extent.

Suppose the training set is $U = \{x_1, x_2, \cdots, x_N\}$, the incremental training set is $E = \{e_1, e_2, \cdots, e_M\}$, and the test set is $T = \{t_1, t_2, \cdots, t_P\}$. The essence of the 3WD-INB incremental feature is to add samples with higher confidence ($\theta$) in the incremental training set to the training set, and to update $P(C_c)$, $P(\overline{C}_c)$, $P\left(v_i^{(h)}\middle|C_c\right)$ and $P\left(v_i^{(h)}\middle|\overline{C}_c\right)$, the process of incremental learning is as follows.

Introduce the confidence level $\theta$. When the confidence level $\theta_j$ of the sample $e_j$ satisfies Equation (15), add the sample to the training set $U$.

$$\theta_j = \max P(C_c) \prod_{i=1}^{n} P\left(v_i^{(h)}\middle|C_c\right), 1 \leq j \leq M \tag{14}$$

$$\theta_j \geq \gamma \sum_{i=1}^{l} \theta_i, 1 \leq l \leq M \tag{15}$$

where $\gamma$ is the confidence coefficient; under normal circumstances $\gamma \in (0.5, \ 1]$.

When the incremental training sample $e_j$ is added to the training set $U$, the updated formulas of $P(C_c)$ and $P(\overline{C}_c)$ are:

$$P(C_c) = \begin{cases} \frac{N+K}{1+N+K}P(C_c), C_b \neq C_c \\ \frac{N+K}{1+N+K}P(C_c) + \frac{1}{1+N+K}, C_b = C_c \end{cases}$$

$$P(\overline{C}_c) = \begin{cases} \frac{N+K}{1+N+K}P(\overline{C}_c), C_b = C_c \\ \frac{N+K}{1+N+K}P(\overline{C}_c) + \frac{1}{1+N+K}, C_b \neq C_c \end{cases} \tag{16}$$

The updated formulas of $P\left(v_i^{(h)}\middle|C_c\right)$ and $P\left(v_i^{(h)}\middle|\overline{C}_c\right)$ are as follows,

$$P\left(v_i^{(h)}\middle|C_c\right) = \begin{cases} \frac{\lambda}{1+\lambda}P\left(v_i^{(h)}\middle|C_c\right), C_b = C_c \wedge v_{ci} \neq v_i^{(h)} \\ \frac{\lambda}{1+\lambda}P\left(v_i^{(h)}\middle|C_c\right) + \frac{1}{1+\lambda}, C_b = C_c \wedge v_{ci} = v_i^{(h)} \\ P\left(v_i^{(h)}\middle|C_c\right), C_b \neq C_c \end{cases}$$

$$P\left(v_i^{(h)}\middle|\overline{C}_c\right) = \begin{cases} \frac{\lambda}{1+\lambda}P\left(v_i^{(h)}\middle|C_c\right), C_b \neq C_c \wedge v_{ci} \neq v_i^{(h)} \\ \frac{\lambda}{1+\lambda}P\left(v_i^{(h)}\middle|C_c\right) + \frac{1}{1+\lambda}, C_b \neq C_c \wedge v_{ci} = v_i^{(h)} \\ P\left(v_i^{(h)}\middle|C_c\right), C_b = C_c \end{cases} \tag{17}$$

The updated formulas for the number of samples and the number of categories are:

$$N = N + 1$$

$$count(C_c) = \begin{cases} count(C_c), C_b \neq C_c \\ count(C_c) + 1, C_b = C_c \end{cases} \tag{18}$$

where $N$ represents the number of samples, $K$ represents the number of categories, $count(C_c)$ represents the number of samples belonging to category $C_c$, $\lambda = |a_i| + count(C_c)$, and $|a_i|$ represents the number of features $a_i$.

*3.3. Classification Rules*

Assume that the parameters after incremental learning are $P(C_c)$, $P(\overline{C}_c)$, $P\left(v_i^{(h)}\middle|C_c\right)$ and $P\left(v_i^{(h)}\middle|\overline{C}_c\right)$. Substituting the naive Bayes classification rule Equation (7) into the three-way decision expected-cost Equation (10) for calculation, since the calculation amount of the continuous addition operation is much smaller than that of the multiplication operation during the operation process, the logarithm of both sides is taken. The method reduces the amount of computation and obtains the minimum cost decision rule as follows:

(P) If there are:

$$R(a_{PC_c}|x_h) \leq R(a_{NC_c}|x_h) \Leftrightarrow \sum_{i=1}^{n} \log \frac{P\left(v_i^{(h)}\middle|C_c\right)}{P\left(v_i^{(h)}\middle|\overline{C}_c\right)} \geq \log \frac{P(\overline{C}_c)}{P(C_c)} + \log \frac{\gamma_{C_c}}{1-\gamma_{C_c}}$$

$$R(a_{PC_c}|x_h) \leq R(a_{BC_c}|x_h) \Leftrightarrow \sum_{i=1}^{n} \log \frac{P\left(v_i^{(h)}\middle|C_c\right)}{P\left(v_i^{(h)}\middle|\overline{C}_c\right)} \geq \log \frac{P(\overline{C}_c)}{P(C_c)} + \log \frac{\alpha_{C_c}}{1-\alpha_{C_c}}$$

Then $x_h \in POS_{(\alpha_{C_c}, \beta_{C_c})}(C_c)$.
(N) If there are:

$$R(a_{NC_c}|x_h) \leq R(a_{PC_c}|x_h) \Leftrightarrow \sum_{i=1}^{n} \log \frac{P\left(v_i^{(h)}\middle|C_c\right)}{P\left(v_i^{(h)}\middle|\overline{C}_c\right)} \geq \log \frac{P(\overline{C}_c)}{P(C_c)} + \log \frac{\gamma_{C_c}}{1-\gamma_{C_c}}$$

$$R(a_{NC_c}|x_h) \leq R(a_{BC_c}|x_h) \Leftrightarrow \sum_{i=1}^{n} \log \frac{P\left(v_i^{(h)}\Big|C_c\right)}{P\left(v_i^{(h)}\Big|\overline{C}_c\right)} \geq \log \frac{P(\overline{C}_c)}{P(C_c)} + \log \frac{\beta_{C_c}}{1 - \beta_{C_c}}$$

Then $x_h \in NEG_{(\alpha_{C_c}, \beta_{C_c})}(C_c)$.

(B) If there are:

$$R(a_{BC_c}|x_h) \leq R(a_{PC_c}|x_h) \Leftrightarrow \sum_{i=1}^{n} \log \frac{P\left(v_i^{(h)}\Big|C_c\right)}{P\left(v_i^{(h)}\Big|\overline{C}_c\right)} \geq \log \frac{P(\overline{C}_c)}{P(C_c)} + \log \frac{\alpha_{C_c}}{1 - \alpha_{C_c}}$$

$$R(a_{BC_c}|x_h) \leq R(a_{NC_c}|x_h) \Leftrightarrow \sum_{i=1}^{n} \log \frac{P\left(v_i^{(h)}\Big|C_c\right)}{P\left(v_i^{(h)}\Big|\overline{C}_c\right)} \geq \log \frac{P(\overline{C}_c)}{P(C_c)} + \log \frac{\beta_{C_c}}{1 - \beta_{C_c}}$$

Then $x_h \in BND_{(\alpha_{C_c}, \beta_{C_c})}(C_c)$.

If let $P = \sum_{i=1}^{n} \log \frac{P\left(v_i^{(h)}\big|C_c\right)}{P\left(v_i^{(h)}\big|\overline{C}_c\right)}$, then obtain a new *POS* domain, NEG domain and BND domain, which is the final 3WD-INB classification rule, as shown in Equation (19).

$$POS_{(\alpha_{C_c}, \beta_{C_c})}(C_c) = \left\{ x_h \Big| P \geq \alpha'_{C_c} \right\}$$
$$NEG_{(\alpha_{C_c}, \beta_{C_c})}(C_c) = \left\{ x_h \Big| P \leq \beta'_{C_c} \right\}$$
$$BND_{(\alpha_{C_c}, \beta_{C_c})}(C_c) = \left\{ x_h \Big| \beta'_{C_c} \leq P \leq \alpha'_{C_c} \right\} \tag{19}$$

where

$$\alpha'_{C_c} = \log \frac{P(\overline{C}_c)}{P(C_c)} + \log \frac{\alpha_{C_c}}{1 - \alpha_{C_c}}$$
$$\beta'_{C_c} = \log \frac{P(\overline{C}_c)}{P(C_c)} + \log \frac{\beta_{C_c}}{1 - \beta_{C_c}} \tag{20}$$

### 3.4. Model Idea

Suppose the training set is $U = \{x_1, x_2, \cdots, x_N\}$, the incremental training set is $E = \{e_1, e_2, \cdots, e_M\}$, and the test set is $T = \{t_1, t_2, \cdots, t_P\}$. First, use the training set $U$ to build a naive Bayes classifier NB and, then, carry out incremental learning according to the incremental training set $E$ to build the INB classifier and, finally, combine the three decision ideas to build the 3WD-IBN classifier to classify the test set $T$. In the algorithm, the maximum number of increments $IM$ is added as a parameter, which can control the process of incremental learning. The general flow of the algorithm is shown in Figure 2.

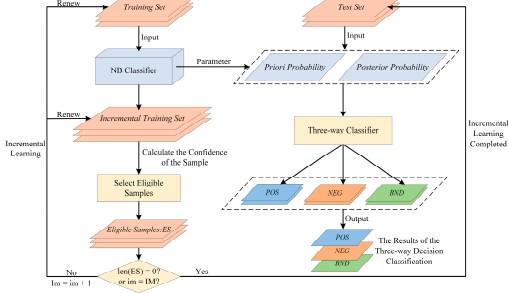

**Figure 2.** 3WD-IBN Algorithm Flow.

The specific operation of the algorithm is as follows (Algorithm 2):

---

**Algorithm 2**: 3WD-INB Classification

---

**Input**: training set $U = \{x_1, x_2, \cdots, x_N\}$; incremental training set $E = \{e_1, e_2, \cdots, e_M\}$; test set $T = \{t_1, t_2, \cdots, t_P\}$; thresholds for each category $(\alpha_{C_c}, \beta_{C_c})$; maximum number of increments $IM$; confidence factor $\gamma$.

**Output**: the three-way classification results of the test set $T$: $\{POS, NEG, BND\}$.

1. Build NB classifier
2. **For** $a_i \in A$ **do**
3. **For** $a_j \in A$ **do**
4. **If** $v_i^{(h)}$ **is** Discrete Data **then**
5. Use (5) (6) to calculate $P(C_c)$, $P(\overline{C}_c)$, $P\left(v_i^{(h)}|C_c\right)$ and $P\left(v_i^{(h)}|\overline{C}_c\right)$
6. **Else**
7. Use (5) to calculate $P(C_c)$, $P(\overline{C}_c)$
8. The optimal distribution obtained by fitting with **Algorithm 1**: $p_i\left(v_i^{(h)}|C_c\right)$, $p_i\left(v_i^{(h)}|\overline{C}_c\right)$
9. $P\left(v_i^{(h)}|C_c\right) = p_i\left(v_i^{(h)}|C_c\right)$, $P\left(v_i^{(h)}|\overline{C}_c\right) = p_i\left(v_i^{(h)}|\overline{C}_c\right)$
10. **End if**
11. **End for**
12. **End for**
13. Conduct an incremental learning process
14. $im = 0$
15. **For** $e_j \in E$ **do**
16. Use (13) to calculate the sample confidence $\theta_j$
17. **If** $\theta_j \geq \gamma \sum_{i=1}^{l} \theta_i, 1 \leq l \leq M$ **then**
18. $E = E - e_j$
19. $U = U + e_j$
20. **If** $v_i^{(h)}$ is discrete data **then**
21. Use (16-18) to update the parameters: $P(C_c)$, $P(\overline{C}_c)$, $P\left(v_i^{(h)}|C_c\right)$, $P\left(v_i^{(h)}|\overline{C}_c\right)$ $count(C_c)$ and $U$
22. **Else**
23. Perform steps 1-8 to reconstruct the NB classifier
24. **End if**
25. **End if**
26. **End for**
27. $im = im + 1$
28. **If** $E = \varnothing$ **or** $im = IM$ **then**
29 Execute 33
30. **Else**
31. Execute 15
32. **End if**
33. Compute the threshold $\left(\alpha'_{C_c}, \beta'_{C_c}\right)$ for category $C_c$ according to (20)
34. Carry out three-way decision-making classification on category $C_c$, and get *POS*, *NEG*, *BND*
35. **For** $t = 1, 2, \cdots, p$ **do**
36. Calculate $P = \sum_{i=1}^{n} \log \dfrac{P\left(v_i^{(h)}|C_c\right)}{P\left(v_i^{(h)}|\overline{C}_c\right)}$
37. **If** $P \geq \alpha'_{C_c}$ **then**
38. $t_t \in POS_{(\alpha_{C_c}, \beta_{C_c})}(C_c)$
39. **Else if** $P \leq \beta'_{C_c}$ **then**
40. $t_t \in NEG_{(\alpha_{C_c}, \beta_{C_c})}(C_c)$
41. **Else if** $\beta'_{C_c} \leq P \leq \alpha'_{C_c}$ **then**
42. $t_t \in BND_{(\alpha_{C_c}, \beta_{C_c})}(C_c)$
43. **End if**
44. **End for**
45. **Return** classification results: $\{POS, NEG, BND\}$

---

## 4. Experimental Results and Analysis

### 4.1. Dataset and Experimental Environment

To verify the classification performance of the algorithm, seven discrete datasets and eight continuous datasets are selected for experiments. The datasets are all from the official website of UCI or Kaggle. The dataset information is shown in Table 3. To ensure that the running environment of the comparison experiment is the same, all the simulation results in this paper are obtained by programming in Python language under the environment of Intel(R) Core(TM) i7-10750H CPU @ 2.60GHz 2.59 GHz, RAM 16GB.

**Table 3.** Dataset Information.

| Name | Type | Number of Samples | Number of Features | Number of Categories |
|---|---|---|---|---|
| Breast | discrete | 699 | 10 | 2 |
| Vote | discrete | 435 | 16 | 2 |
| Mushroom | discrete | 8124 | 22 | 2 |
| Chess | discrete | 3196 | 36 | 2 |
| Hayes-Roth | discrete | 160 | 5 | 3 |
| Car Evaluation | discrete | 1728 | 6 | 4 |
| Lymphography | discrete | 148 | 18 | 4 |
| WDBC | continuous | 569 | 30 | 2 |
| Pima Indians Diabetes | continuous | 766 | 9 | 2 |
| Banknote Authentication | continuous | 1372 | 5 | 2 |
| Magic04 | continuous | 19,020 | 11 | 2 |
| Iris | continuous | 150 | 4 | 3 |
| Waveform | continuous | 5000 | 22 | 3 |
| Glass | continuous | 214 | 9 | 6 |
| Segmentation | continuous | 2310 | 19 | 7 |

### 4.2. Evaluation Indicators

For the traditional binary classification model of binary decision-making, accuracy (*ACC*), recall (*Recall*), precision (*Precision*), and F1-score (*F*1) are usually used to evaluate the classification performance. These indicators are based on the classification confusion matrix of binary decision-making, as shown in Table 4.

**Table 4.** Two-way decision confusion matrix.

| Reference | Prediction | |
|---|---|---|
| | **Positive** | **Negative** |
| Positive | *TP* | *FN* |
| Negative | *FP* | *TN* |

Accuracy (*ACC*) describes the overall classification accuracy, as shown in Equation (21).

$$ACC = \frac{TP + TN}{TP + TN + FP + FN} \tag{21}$$

*Recall* is the ability of the classifier to find all positive samples. The *Recall* value is 1 at best and 0 at worst. The calculation process is Equation (22).

$$Recall = \frac{TP}{TP + FN} \tag{22}$$

*Precision* is the ability of the classifier not to label negative examples as positive examples. The best value of *Precision* is 1, and the worst is 0. The calculation process is Equation (23).

$$Precision = \frac{TP}{TP + FP} \tag{23}$$

F1-score (*F1*) can be regarded as a harmonic average of model precision and recall, with a maximum value of 1 and a minimum value of 0. The calculation formula is shown in Equation (24).

$$F1 = \frac{2 \times ACC \times Recall}{ACC + Recall} \tag{24}$$

Since F1-score can comprehensively reflect the accuracy rate (*ACC*) and recall rate (*Recall*), this paper selects two indicators of F1-score (*F1*) and *Precision* to evaluate the classification performance.

For the evaluation index of the three-way decision model, this paper refers to the processing method of Jia et al. [43]. The classification confusion matrix of the three decisions is shown in Table 5. In Table 5, $n_{xy}$ represents the number of samples when the actual class *x* is judged as class *y*.

**Table 5.** Three-way decision confusion matrix.

|  | Actual Positive Domain | Actual Negative Domain |
|---|:---:|:---:|
| Predicted as *POS* domain | $n_{PP}$ | $n_{PN}$ |
| Predicted as *BND* domain | $n_{BP}$ | $n_{BN}$ |
| Predicted as *NEG* domain | $n_{NP}$ | $n_{NN}$ |

For the accuracy of the three-way decision, the calculation formula is shown in Equation (25).

$$ACC = \frac{n_{PP} + n_{NN}}{n_{PP} + n_{PN} + n_{NP} + n_{NN}} \tag{25}$$

For *Recall*, it is necessary to take into account the positive samples divided into the *NEG* domain, and the calculation formula is shown in Equation (26).

$$Recall = \frac{n_{PP}}{n_{PP} + n_{NP} + n_{NP}} \tag{26}$$

For *Precision*, it is necessary to take into account the positive samples divided into the *NEG* domain, and the calculation formula is shown in Equation (27).

$$Precision = \frac{n_{PP}}{n_{PP} + n_{PN} + n_{NP}} \tag{27}$$

The calculation formula of the F1-score is consistent with Equation (24).

### 4.3. Parameter Changes and Analysis of Experimental Results

Use the dataset to use 3WD-INB for classification testing, using fivefold cross-validation, four for the training set, one for the test set, where the ratio of the training set in the training set to the incremental training set is $U : E = 1 : 3$, and the maximum number of increments $IM = 20$. Use *F1* and *Precision* to evaluate results.

#### 4.3.1. Threshold $\alpha$ and $\beta$ Change Analysis

Thresholds $\alpha$ and $\beta$ are important parameters of the 3WD-INB model, usually between 0 and 1. In the experiment of exploring the change of thresholds $\alpha$ and $\beta$, keep the confidence coefficient $\gamma = 0.7$ unchanged, and the thresholds $\alpha$ and $\beta$ take a fixed step 0.1 change for experiments.

For discrete data, take the mushroom and breast datasets as examples, and for continuous data, take the WDBC dataset as an example. Figures 3–8 show the change of *F1* and *Precision* as the threshold $(\alpha, \beta)$ changes for the three datasets (only the results of one category of each dataset are shown).

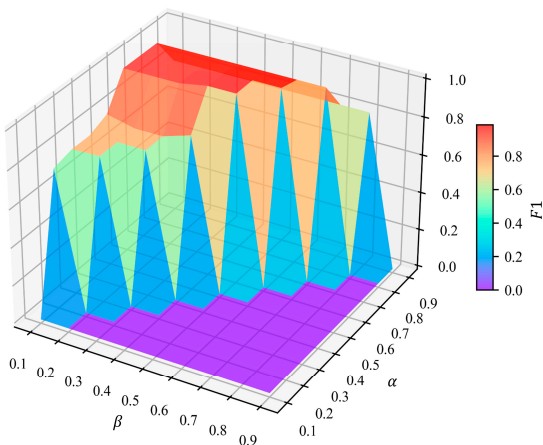

**Figure 3.** Breast's *F*1 change chart.

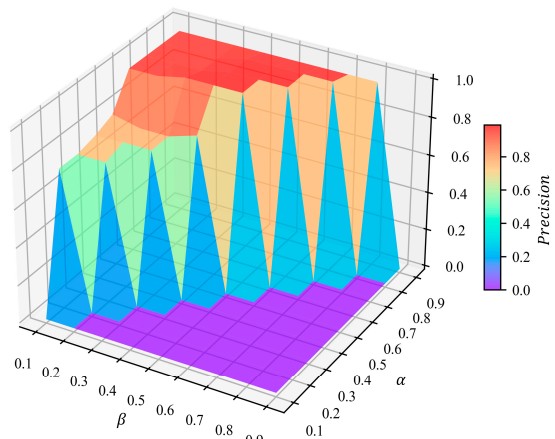

**Figure 4.** Breast's *Precision* change chart.

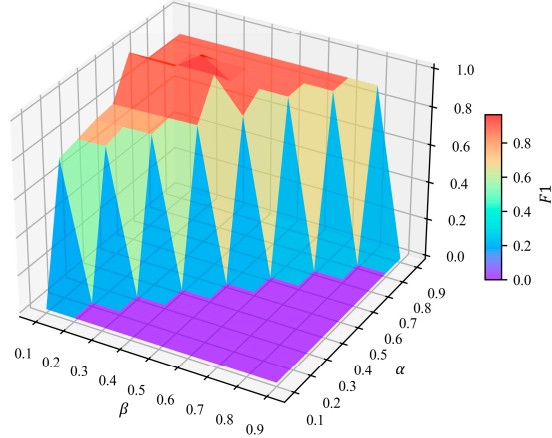

**Figure 5.** Mushroom's *F*1 change chart.

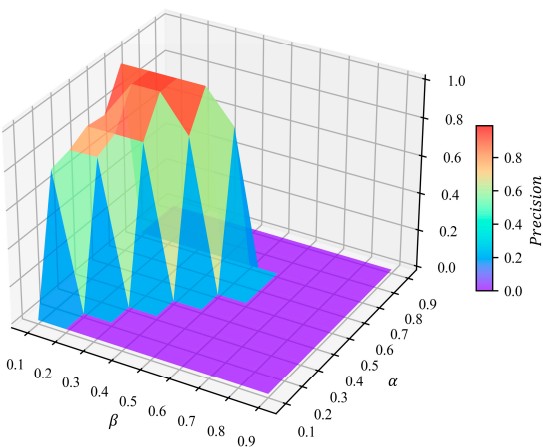

**Figure 6.** Mushroom's *Precision* change chart.

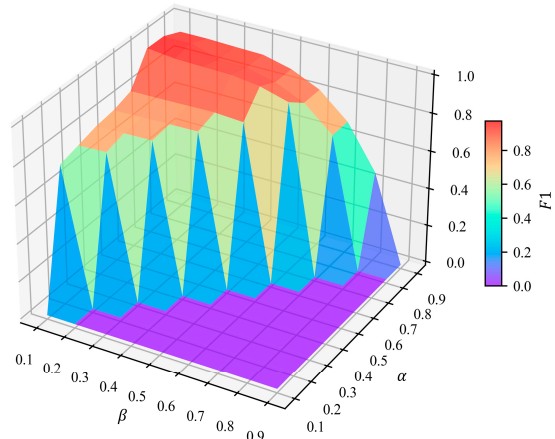

**Figure 7.** WDBC's *F*1 change chart.

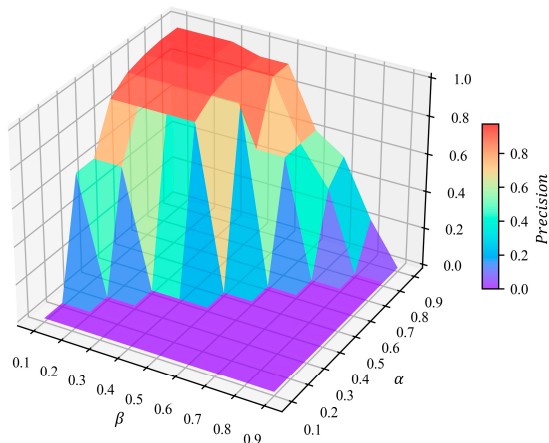

**Figure 8.** WDBC's *Precision* change chart.

It can be seen from the figure that as the threshold value $(\alpha, \beta)$ changes, the evaluation indicators *F*1 and *Precision* are also constantly changing, achieving better-expected results. Due to the regulation of $\alpha > \beta$, in the case of $\alpha \leq \beta$, the values of *F*1 and *Precision* are both 0; in the case of $\alpha > \beta$, the overall *F*1 and *Precision* of the classification results of 3WD-INB have maintained a relatively high level; due to the comprehensive, consider *F*1 and *Precision*, so using the average of the two as a reference, the breast dataset is optimal

at $\alpha = 0.7, \beta = 0.6$, the mushroom dataset is optimal at $\alpha = 0.6, \;\; \beta = 0.3$, and WDBC is optimal at $\alpha = 0.8, \;\; \beta = 0.2$.

Simulation experiments are also carried out on other datasets, and the optimal threshold combination $(\alpha, \beta)$ under each category of all selected datasets is obtained, as shown in Table 6.

**Table 6.** Optimal Threshold.

| Dataset Name | Category | | | | | | |
|---|---|---|---|---|---|---|---|
| | $C_1$ | $C_2$ | $C_3$ | $C_4$ | $C_5$ | $C_6$ | $C_7$ |
| Breast | (0.7, 0.6) | (0.5, 0.2) | - | - | - | - | - |
| Vote | (0.4, 0.1) | (0.4, 0.1) | - | - | - | - | - |
| Mushroom | (0.6, 0.3) | (0.2, 0.1) | - | - | - | - | - |
| Chess | (0.4, 0.3) | (0.5, 0.2) | - | - | - | - | - |
| Hayes-Roth | (0.4, 0.3) | (0.4, 0.3) | (0.5, 0.4) | - | - | - | - |
| Car Evaluation | (0.5, 0.3) | (0.4, 0.3) | (0.7, 0.6) | (0.4, 0.3) | - | - | - |
| Lymphography | (0.3, 0.2) | (0.5, 0.3) | (0.5, 0.4) | (0.8, 0.6) | - | - | - |
| WDBC | (0.8, 0.2) | (0.5, 0.4) | - | - | - | - | - |
| Pima Indians Diabetes | (0.3, 0.1) | (0.2, 0.1) | - | - | - | - | - |
| Banknote Authentication | (0.6, 0.1) | (0.2, 0.1) | - | - | - | - | - |
| Magic04 | (0.3, 0.2) | (0.6, 0.5) | - | - | - | - | - |
| Iris | (0.5, 0.4) | (0.4, 0.3) | (0.4, 0.3) | - | - | - | - |
| Waveform | (0.2, 0.1) | (0.5, 0.1) | (0.2, 0.1) | - | - | - | - |
| Glass | (0.5, 0.3) | (0.4, 0.3) | (0.2, 0.1) | (0.4, 0.3) | (0.2, 0.1) | (0.4, 0.3) | - |
| Segmentation | (0.6, 0.5) | (0.8, 0.5) | (0.2, 0.1) | (0.4, 0.3) | (0.3, 0.2) | (0.5, 0.4) | (0.5, 0.4) |

4.3.2. Change Analysis of Confidence Coefficient $\gamma$

The confidence coefficient $\gamma$ is an important parameter in the incremental learning process, generally between 0.5 and 1. In the exploration of the change analysis experiment of confidence coefficient $\gamma$, the optimal threshold $(\alpha, \beta)$ in Table 6 is used for each dataset. The maximum number of increments $IM = 20$, and the confidence coefficient $\gamma$ is tested with a step size of 0.05.

After experiments, the $F1$ changes of all datasets are shown in Table 7 (only one of the changes is shown).

In the analysis, confidence factor $\gamma$ plays a good role in the incremental learning stage. From the test results of the experiment, in most cases, with the increase in the confidence factor $\gamma$, the value of $F1$ usually rises first and then decreases, and the highest point tends to appear in the middle. For breast datasets, the peak locations occur between 0.65 and 0.8, indicating that the confidence factor $\gamma$ is the best set in the next range. From the analysis, we can see that the WDBC and iris datasets are not aware of the confidence factor $\gamma$ and have a wide optimal confidence range, which may be due to the high quality of the dataset. The breast and chess datasets show that when the confidence coefficient $\gamma$ increases from 0.55 to 0.95, the corresponding classification accuracy first remains unchanged from the maximum value, then decreases gradually. The WDBC dataset is insensitive to the confidence factor $\gamma$, and the accuracy remains unchanged at 0.9735. The vote and other datasets show a trend of change that first rises and then decreases. Therefore, it is concluded that the larger the confidence factor $\gamma$, the higher the data quality requirements for incremental training samples of incremental learning, but the larger the $\gamma$, the better, because a larger confidence factor $\gamma$ may lose a large number of features in incremental training set samples. The smaller the confidence factor $\gamma$, the lower the data quality requirement for incremental training samples, and a large number of features can be preserved, but the poorer samples may also be added to the training set. Therefore, in the process of practical application, $\gamma$ can be adjusted reasonably according to the data quality, and good results can be achieved.

**Table 7.** Accuracy Statistics Table.

| Dataset Name | Confidence Coefficient $\gamma$ | | | | | | | | |
|---|---|---|---|---|---|---|---|---|---|
| | 0.55 | 0.6 | 0.65 | 0.7 | 0.75 | 0.8 | 0.85 | 0.9 | 0.95 |
| Breast | 0.9015 | 0.9559 | 0.9856 | 0.9856 | 0.9856 | 0.9856 | 0.9715 | 0.9665 | 0.9665 |
| Vote | 0.9654 | 0.9786 | 0.9786 | 0.9786 | 0.9786 | 0.9551 | 0.9551 | 0.9441 | 0.9441 |
| Mushroom | 0.9775 | 0.9775 | 0.9898 | 0.9898 | 0.9898 | 0.9625 | 0.9625 | 0.9625 | 0.9459 |
| Chess | 0.8975 | 0.9001 | 0.9010 | 0.9115 | 0.9115 | 0.9115 | 0.9059 | 0.8949 | 0.8516 |
| Hayes-Roth | 0.9010 | 0.9103 | 0.8915 | 0.8915 | 0.8915 | 0.8814 | 0.8810 | 0.8810 | 0.8810 |
| Car Evaluation | 0.9809 | 0.9809 | 0.9809 | 0.9809 | 0.9755 | 0.9713 | 0.9215 | 0.9359 | 0.9001 |
| Lymphography | 0.8715 | 0.8990 | 0.8990 | 0.8850 | 0.8850 | 0.8850 | 0.8211 | 0.8013 | 0.8001 |
| WDBC | 0.9851 | 0.9851 | 0.9851 | 0.9851 | 0.9851 | 0.9851 | 0.9801 | 0.9799 | 0.9799 |
| Pima Indians Diabetes | 0.8859 | 0.8859 | 0.8919 | 0.8990 | 0.9015 | 0.9015 | 0.8915 | 0.8915 | 0.8126 |
| Banknote Authentication | 0.8915 | 0.8891 | 0.8891 | 0.9167 | 0.9167 | 0.9167 | 0.9011 | 0.9011 | 0.8756 |
| Magic04 | 0.9557 | 0.9557 | 0.9557 | 0.9557 | 0.9355 | 0.9355 | 0.9215 | 0.9200 | 0.9167 |
| Iris | 0.9875 | 1.0000 | 1.0000 | 1.0000 | 1.0000 | 1.0000 | 1.0000 | 1.0000 | 0.9810 |
| Waveform | 0.8210 | 0.8390 | 0.8390 | 0.8220 | 0.8150 | 0.8001 | 0.7995 | 0.7995 | 0.7551 |
| Glass | 0.6655 | 0.6655 | 0.6744 | 0.6744 | 0.6744 | 0.6679 | 0.6679 | 0.6679 | 0.6679 |
| Segmentation | 0.9755 | 0.9815 | 0.9815 | 0.9815 | 0.9815 | 0.9713 | 0.9703 | 0.9667 | 0.9667 |

*4.4. Comparative Experimental Analysis*

4.4.1. Comparative Analysis in Discrete Data

Using discrete datasets, use 3WD-INB for classification testing, select RF, SVM, KNN, NB, INB, and NB-IPCA (from the literature [2]) for comparative experiments and use fivefold cross-validation, four for the training set, one for the test set, where the ratio of the training set in the training set to the incremental training set is $U : E = 1 : 3$, and *F1* and *Recall* are used for the result evaluation. After the experiment, the results shown in Table 8 are obtained (showing the evaluation indicators of all categories).

As can be seen from Table 8, when the threshold $(\alpha, \beta)$ and confidence factor $\gamma$ are given a certain value, the classification performance of 3WD-INB is better than other comparable models in most cases. For the mushroom datasets with a large number of samples, the results are the best, except that the $C_2$ category is lower than the SVM model. For the chess datasets with more features, the results of 3WD-INB are higher than those of comparable models. For the Hayes-Roth datasets with a small number of features and samples, the results of 3WD-INB are also significantly superior, with the $C_3$ class having better performance in different classifiers. For the breast and vote datasets with a moderate number of samples and features, 3WD-INB has a significantly better effect on breast than other models and has a smaller difference in the $C_1$ category of vote than the RF and SVM models. For the car and lymphography datasets with a large number of categories, the overall classification performance of 3WD-INB is much better than other types of Bayesian models and traditional models. After the above analysis, 3WD-INB has a good effect on datasets with different numbers of samples and features, as well as on the learning of multiple classified samples, but overall, for datasets with a higher number of features, although compared with other models, there is still room for improvement objectively.

4.4.2. Comparative Analysis in Continuous Data

Using discrete datasets, 3WD-INB was used for classification testing, RF, SVM, MLP, D-NB, and G-NB were selected for comparative experiments, fivefold cross-validation was adopted, four were the training sets, and one was the testing set. The ratio of the training set is $U : E = 1 : 3$, and *F1* and *Recall* are used for the result evaluation. After the experiment, the results shown in Table 9 are obtained (showing the evaluation indicators of all categories).

**Table 8.** 3WD-INB Experimental Result 1 (bold is the best result).

| Dataset | Algorithm | $C_1$ | | $C_2$ | | $C_3$ | | $C_4$ | |
|---|---|---|---|---|---|---|---|---|---|
| | | F1 | Precision | F1 | Precision | F1 | Precision | F1 | Precision |
| Breast | RF | 0.9282 | 0.9333 | 0.8687 | 0.8600 | - | - | - | - |
| | SVM | 0.9432 | 0.9765 | 0.9038 | 0.8545 | - | - | - | - |
| | KNN | 0.9333 | 0.9438 | 0.8800 | 0.8627 | - | - | - | - |
| | NB | 0.9125 | 0.9044 | 0.9001 | 0.8774 | - | - | - | - |
| | INB | 0.9545 | 0.9655 | 0.9231 | 0.8728 | - | - | - | - |
| | NB-IPCA | 0.9545 | 0.9655 | 0.9231 | 0.8728 | - | - | - | - |
| | 3WD-INB | **0.9856** | **0.9882** | **0.9531** | **0.9327** | - | - | - | - |
| Vote | RF | **0.9789** | **0.9819** | 0.9661 | 0.9554 | - | - | - | - |
| | SVM | **0.9789** | **0.9819** | 0.9514 | 0.9667 | - | - | - | - |
| | KNN | 0.9663 | 0.9556 | 0.9647 | 0.9762 | - | - | - | - |
| | NB | 0.9351 | 0.9115 | 0.9356 | 0.9222 | - | - | - | - |
| | INB | 0.9535 | 0.9762 | 0.9545 | 0.9333 | - | - | - | - |
| | NB-IPCA | 0.9655 | 0.9767 | 0.9655 | 0.9545 | - | - | - | - |
| | 3WD-INB | 0.9786 | 0.9815 | **0.9751** | **0.9799** | - | - | - | - |
| Mushroom | RF | 0.9811 | 0.9885 | 0.9781 | 0.9711 | - | - | - | - |
| | SVM | 0.9615 | 0.9789 | 0.9893 | **1.0000** | - | - | - | - |
| | KNN | 0.9551 | 0.9245 | 0.9333 | 0.9125 | - | - | - | - |
| | NB | 0.9505 | 0.9145 | 0.9402 | 0.9872 | - | - | - | - |
| | INB | 0.9516 | 0.9156 | 0.9416 | 0.9886 | - | - | - | - |
| | NB-IPCA | 0.9654 | 0.9199 | 0.9215 | 0.9335 | - | - | - | - |
| | 3WD-INB | **0.9898** | **0.9891** | **0.9994** | 0.9981 | - | - | - | - |
| Chess | RF | 0.9005 | 0.9292 | 0.8849 | 0.9149 | - | - | - | - |
| | SVM | 0.8999 | 0.8999 | 0.8900 | 0.8855 | - | - | - | - |
| | KNN | 0.8848 | 0.9356 | 0.8849 | 0.8749 | - | - | - | - |
| | NB | 0.8950 | 0.9226 | 0.8749 | 0.9001 | - | - | - | - |
| | INB | 0.9005 | 0.9292 | 0.8887 | 0.9015 | - | - | - | - |
| | NB-IPCA | 0.9005 | 0.9335 | 0.8955 | 0.9119 | - | - | - | - |
| | 3WD-INB | **0.9115** | **0.9433** | **0.8987** | **0.9211** | - | - | - | - |
| Hayes-Roth | RF | 0.8387 | 0.9285 | 0.7368 | 0.6364 | **1.0000** | **1.0000** | - | - |
| | SVM | 0.7407 | 1.0000 | 0.6957 | 0.5333 | **1.0000** | **1.0000** | - | - |
| | KNN | 0.6842 | 0.6190 | 0.3529 | 0.3333 | 0.4444 | **1.0000** | - | - |
| | NB | 0.8485 | 0.8750 | 0.7778 | 0.7000 | 0.9231 | **1.0000** | - | - |
| | INB | 0.8387 | 0.9286 | 0.7368 | 0.6364 | **1.0000** | **1.0000** | - | - |
| | NB-IPCA | 0.8559 | 0.9359 | 0.8005 | 0.8115 | 0.9545 | 0.9887 | - | - |
| | 3WD-INB | **0.9103** | **1.0000** | **0.8551** | **0.8445** | **1.0000** | **1.0000** | - | - |
| Car Evaluation | RF | 0.9682 | 0.9500 | 0.8182 | **1.0000** | 0.9979 | 0.9559 | **1.0000** | 1.0000 |
| | SVM | 0.8322 | 0.8611 | 0.6667 | 0.8750 | 0.9697 | 0.9449 | 0.8889 | **1.0000** |
| | KNN | 0.9487 | 0.9367 | 0.8182 | **1.0000** | 0.9938 | 0.9917 | 0.9677 | 0.9375 |
| | NB | 0.7917 | 0.8507 | 0.6364 | 0.7778 | 0.9539 | 0.9225 | 0.8889 | **1.0000** |
| | INB | 0.9557 | 0.8915 | 0.8855 | 0.9225 | 0.9656 | 0.9115 | 0.9005 | 0.9375 |
| | NB-IPCA | 0.7273 | 0.7273 | 0.4000 | 0.5714 | 0.9535 | 0.9291 | 0.6957 | **1.0000** |
| | 3WD-INB | **0.9809** | **0.9625** | **0.9167** | **1.0000** | **1.0000** | **1.0000** | 0.9655 | **1.0000** |
| Lymphography | RF | **0.8990** | **1.0000** | 0.8485 | 0.7778 | 0.8000 | 0.8333 | **1.0000** | **1.0000** |
| | SVM | **0.8990** | **1.0000** | 0.8485 | 0.7778 | 0.8000 | 0.8333 | **1.0000** | **1.0000** |
| | KNN | **0.8990** | **1.0000** | 0.8965 | 0.9286 | 0.8571 | 0.8000 | **1.0000** | **1.0000** |
| | NB | 0.8080 | **1.0000** | 0.8550 | 0.7946 | 0.8551 | 0.8657 | **1.0000** | **1.0000** |
| | INB | 0.8152 | **1.0000** | 0.9286 | **1.0000** | 0.8966 | 0.8125 | **1.0000** | **1.0000** |
| | NB-IPCA | 0.8556 | **1.0000** | 0.9116 | 0.9445 | 0.8988 | 0.8449 | **1.0000** | **1.0000** |
| | 3WD-INB | **0.8990** | **1.0000** | **0.9333** | 0.9333 | **0.8989** | **0.8571** | **1.0000** | **1.0000** |

**Table 9.** 3WD-INB Experimental Result 2 (bold is the best result).

| Dataset | Algorithm | $c_1$ F1 | $c_1$ Precision | $c_2$ F1 | $c_2$ Precision | $c_3$ F1 | $c_3$ Precision | $c_4$ F1 | $c_4$ Precision |
|---|---|---|---|---|---|---|---|---|---|
| WDBC | RF | 0.9559 | 0.9285 | 0.9348 | 0.9773 | - | - | - | - |
| | SVM | 0.9103 | 0.8354 | 0.8433 | **1.0000** | - | - | - | - |
| | MLP | 0.9231 | 0.8571 | 0.8705 | **1.0000** | - | - | - | - |
| | D-NB | 0.9736 | 0.9659 | 0.9636 | 0.9302 | - | - | - | - |
| | G-NB | 0.9386 | 0.9545 | 0.9386 | 0.9167 | - | - | - | - |
| | 3WD-INB | **0.9851** | **0.9667** | **0.9736** | **1.0000** | - | - | - | - |
| Pima Indians Diabetes | RF | 0.8186 | 0.7333 | 0.6806 | 0.7941 | - | - | - | - |
| | SVM | 0.8089 | 0.7000 | 0.5819 | 0.8333 | - | - | - | - |
| | MLP | 0.8059 | 0.7652 | 0.7355 | 0.7083 | - | - | - | - |
| | D-NB | 0.7662 | 0.7979 | 0.7662 | 0.7091 | - | - | - | - |
| | G-NB | 0.8403 | 0.8632 | 0.7403 | **0.8424** | - | - | - | - |
| | 3WD-INB | **0.9015** | **0.8891** | **0.8551** | 0.8312 | - | - | - | - |
| Banknote Authentication | RF | **0.9933** | **1.0000** | 0.9545 | 0.9774 | - | - | - | - |
| | SVM | 0.9866 | 0.9795 | 0.9959 | 0.9779 | - | - | - | - |
| | MLP | **0.9933** | **1.0000** | 0.8551 | 0.8059 | - | - | - | - |
| | D-NB | 0.8073 | 0.8986 | 0.8073 | 0.7008 | - | - | - | - |
| | G-NB | 0.8036 | 0.8553 | 0.8036 | 0.7398 | - | - | - | - |
| | 3WD-INB | 0.9167 | 0.9551 | **0.9967** | **0.9840** | - | - | - | - |
| Magic04 | RF | 0.9079 | 0.8780 | 0.8134 | 0.8743 | - | - | - | - |
| | SVM | 0.8724 | 0.8062 | 0.6957 | 0.8650 | - | - | - | - |
| | MLP | 0.8684 | 0.8436 | 0.7373 | 0.7833 | - | - | - | - |
| | D-NB | 0.7559 | 0.8059 | 0.9335 | 0.9011 | - | - | - | - |
| | G-NB | 0.9204 | 0.9115 | 0.8995 | **0.9559** | - | - | - | - |
| | 3WD-INB | **0.9557** | **0.9226** | **0.9458** | 0.9175 | - | - | - | - |
| Iris | RF | **1.0000** | **1.0000** | **1.0000** | **1.0000** | **1.0000** | **1.0000** | - | - |
| | SVM | **1.0000** | **1.0000** | **1.0000** | **1.0000** | 0.9454 | 0.9285 | - | - |
| | MLP | 0.9559 | 0.9885 | **1.0000** | **1.0000** | **1.0000** | **1.0000** | - | - |
| | D-NB | **1.0000** | **1.0000** | **1.0000** | 0.9333 | **1.0000** | 0.7778 | - | - |
| | G-NB | **1.0000** | **1.0000** | **1.0000** | **1.0000** | **1.0000** | **1.0000** | - | - |
| | 3WD-INB | **1.0000** | **1.0000** | **1.0000** | **1.0000** | **1.0000** | **1.0000** | - | - |
| Waveform | RF | 0.8253 | 0.8435 | 0.8797 | 0.8746 | 0.8787 | 0.8676 | - | - |
| | SVM | 0.8152 | 0.8673 | 0.8673 | 0.8588 | 0.8571 | 0.8559 | - | - |
| | MLP | 0.7926 | 0.8144 | 0.8526 | 0.8551 | 0.8760 | 0.8543 | - | - |
| | D-NB | 0.8090 | 0.4984 | 0.8810 | 0.8452 | 0.8840 | 0.9277 | - | - |
| | G-NB | **0.8390** | 0.5285 | 0.8780 | 0.9020 | 0.8870 | 0.8995 | - | - |
| | 3WD-INB | **0.8390** | **0.8850** | **0.8991** | **0.9105** | **0.9115** | **0.9456** | - | - |

**Glass**

| Algorithm | $c_1$ F1 | $c_1$ Precision | $c_2$ F1 | $c_2$ Precision | $c_3$ F1 | $c_3$ Precision | $c_4$ F1 | $c_4$ Precision |
|---|---|---|---|---|---|---|---|---|
| RF | 0.7115 | 0.7335 | 0.6667 | 0.5998 | 0.6159 | 0.6667 | 0.8559 | 0.9556 |
| SVM | 0.7200 | 0.7500 | 0.7500 | 0.8000 | 0.5000 | 0.6510 | **1.0000** | **1.0000** |
| MLP | **0.7428** | 0.7222 | 0.7407 | 0.8333 | 0.5000 | 0.4286 | 0.6667 | **1.0000** |
| D-NB | 0.6744 | 0.6364 | 0.6512 | 0.2143 | 0.6233 | 0.6667 | 0.9069 | 0.2500 |
| G-NB | 0.7097 | 0.7557 | 0.6667 | 0.5882 | 0.3333 | 0.5000 | 0.6667 | **1.0000** |
| 3WD-INB | 0.7244 | **0.7647** | **0.7857** | **0.8462** | **0.6520** | **0.6995** | 0.6667 | **1.0000** |

| Algorithm | $c_5$ F1 | $c_5$ Precision | $c_6$ F1 | $c_6$ Precision | - | - | - | - |
|---|---|---|---|---|---|---|---|---|
| RF | 0.7500 | 0.5656 | 0.6519 | 0.7500 | - | - | - | - |
| SVM | 0.8000 | **1.0000** | 0.6667 | 0.6667 | - | - | - | - |
| MLP | 0.6667 | 0.5000 | 0.6667 | 0.6667 | - | - | - | - |
| D-NB | 0.6667 | 0.5000 | 0.6667 | 0.6667 | - | - | - | - |
| G-NB | 0.7571 | 0.6556 | 0.6667 | 0.6667 | - | - | - | - |

**Segmentation**

| Algorithm | $c_1$ F1 | $c_1$ Precision | $c_2$ F1 | $c_2$ Precision | $c_3$ F1 | $c_3$ Precision | $c_4$ F1 | $c_4$ Precision |
|---|---|---|---|---|---|---|---|---|
| RF | 0.9800 | 0.9608 | 0.8595 | 0.9455 | 0.9185 | 0.9118 | **1.0000** | **1.0000** |
| SVM | 0.8085 | 0.9268 | 0.8955 | 0.8955 | 0.6713 | 0.6575 | **1.0000** | **1.0000** |
| MLP | **0.9906** | 0.9814 | 0.9403 | 0.9403 | 0.9067 | 0.8500 | **1.0000** | **1.0000** |
| D-NB | 0.9500 | 0.9245 | 0.9452 | 0.8656 | 0.8310 | 0.1429 | **1.0000** | **1.0000** |
| G-NB | 0.9548 | 0.9592 | 0.9338 | 0.8788 | 0.9554 | 0.6779 | **1.0000** | **1.0000** |
| 3WD-INB | 0.9815 | **0.9849** | 0.9466 | 0.9688 | **0.9744** | 0.9500 | **1.0000** | **1.0000** |

| Algorithm | $c_5$ F1 | $c_5$ Precision | $c_6$ F1 | $c_6$ Precision | $c_7$ F1 | $c_7$ Precision | - | - |
|---|---|---|---|---|---|---|---|---|
| RF | 0.9752 | 0.9516 | **1.0000** | **1.0000** | 0.8264 | 0.7937 | - | - |
| SVM | 0.9925 | 0.9850 | **1.0000** | **1.0000** | 0.5873 | 0.5522 | - | - |
| MLP | 0.9778 | 0.9865 | **1.0000** | **1.0000** | 0.8462 | **0.9778** | - | - |
| D-NB | 0.9833 | 0.9242 | **1.0000** | **1.0000** | 0.8000 | 0.5763 | - | - |
| G-NB | 0.9976 | 0.9795 | **1.0000** | **1.0000** | 0.8238 | 0.7414 | - | - |
| 3WD-INB | **1.0000** | **1.0000** | **1.0000** | **1.0000** | **0.9120** | 0.9048 | - | - |

From Table 9, it can be concluded that 3WD-INB performs better than other comparable models in most cases, even for continuous data, when the threshold AB and the confidence factor Y are given a certain value. For the large Magic04 dataset, 3WD-INB gives full play to the advantages that Bayesian should have. Classification performance is obviously lower than the G-NB classifier only under the CC class because of the traditional model. For the waveform datasets with the same number of samples, the results of each index 3WD-INB under the three types of results are better, and the RF model is also better. For the WDBC datasets with a large number of features, 3WD-INB is optimal in all categories. For the iris datasets with a small number of features and samples, 3WD-INB meets the criteria of a perfect classifier, which proves that 3WD-INB has excellent performance with a low number of features. For the Pima Indians diabetes and banknote authentication datasets with a moderate number of samples and features, 3WD-INB performs better and more stably than traditional classifiers and has greater overall advantages, and all indicators are improved to a great extent. For the multi-classification datasets of glass and segmentation,

3WD-INB also showed satisfactory results. MLP and SVM are only better than 3WD-INB in individual categories, and the 3WD-INB classifier is more stable. Overall, 3WD-INB is also stable on continuous datasets, regardless of whether it is a two-class or multi-class task.

### 4.4.3. Algorithm Time-Consumption Analysis

The algorithm's time consumption is an important index to evaluate an algorithm. We expect that the algorithm will still perform well with lower time consumption. Due to the different configurations of different running environments, the direct running time does not accurately reflect the time consumption of the algorithm, so we select the fastest-running algorithm as the benchmark algorithm, make its running time 1, and then test the relative running time of other algorithms and the base algorithm. For discrete data, select the fastest NB classifier as the base algorithm. For continuous data, G-NB is chosen as the base algorithm.

After testing, the running time of each algorithm under discrete data is shown in Table 10, and the running time of each algorithm under continuous data is shown in Table 11.

**Table 10.** Algorithm time consumption under discrete data.

| Dataset Name | The Running Time of the Relative Basis Algorithm | | | | | | |
|---|---|---|---|---|---|---|---|
| | **RF** | **SVM** | **KNN** | **NB** | **INB** | **NB-IPCA** | **3WD-INB** |
| Breast | 19.4 | 2.5 | 2.7 | 1 | 1.2 | 3.9 | 1.9 |
| Vote | 18.9 | 1.1 | 1.7 | 1 | 1.6 | 2.6 | 1.4 |
| Mushroom | 21.2 | 63.8 | 36.0 | 1 | 11.6 | 15.9 | 12.5 |
| Chess | 18.2 | 19.8 | 4.5 | 1 | 2.7 | 9.5 | 2.8 |
| Hayes-Roth | 15.1 | 1.1 | 1.5 | 1 | 1.5 | 1.6 | 1.9 |
| Car Evaluation | 18.8 | 14.0 | 2.7 | 1 | 3.3 | 4.6 | 3.3 |
| Lymphography | 13.1 | 1.1 | 1.1 | 1 | 1.5 | 2.5 | 1.3 |

**Table 11.** Algorithm time consumption under continuous data.

| Dataset Name | The Running Time of the Relative Basis Algorithm | | | | | |
|---|---|---|---|---|---|---|
| | **RF** | **SVM** | **MLP** | **D-NB** | **G-NB** | **3WD-INB** |
| WDBC | 37.1 | 2.5 | 25.8 | 1.3 | 1 | 11.0 |
| Pima Indians Diabetes | 27.0 | 5.0 | 227.3 | 1.8 | 1 | 6.1 |
| Banknote Authentication | 52.4 | 3.6 | 302.8 | 2.9 | 1 | 8.0 |
| Magic04 | 214.1 | 360.4 | 190.3 | 35.5 | 1 | 52.5 |
| Iris | 26.6 | 1.3 | 23.4 | 1.2 | 1 | 5.4 |
| Waveform | 182.8 | 134.0 | 581.7 | 6.5 | 1 | 75.3 |
| Glass | 17.5 | 1.7 | 4.8 | 1.5 | 1 | 6.4 |
| Segmentation | 44.6 | 7.8 | 488.0 | 1.9 | 1 | 8.3 |

After analysis, it can be seen that under the discrete data, since 3WD-INB does not need distribution fitting, the time consumption of the algorithm is close to that of the NB classifier. Compared with the traditional RF, SVM and other algorithms, the time consumption is shorter under the same conditions. Under continuous data, since 3WD-INB needs to fit the data distribution, the time-consuming performance of the algorithm is not as good as that under discrete data, but the overall time consumption is still due to the RF and MLP models. To sum up, 3WD-INB is not bad in terms of algorithm time consumption. In most cases, the time consumption is relatively low, and it may decrease when the number of attributes is large.

## 5. Conclusions and Future Work

Considering that in the process of classification, uncertain objects are forcibly divided into certain categories that do not conform to people's actual decision-making processes and

real-world data are often acquired dynamically; combining incremental learning, three-way decision ideas, and naive Bayes classifiers, a three-way incremental naive Bayes classifier (3WN-INB) is proposed. Screen samples with high data quality through incremental learning, perform three-way classification through three-way decision thinking, and use distribution fitting for continuous data to estimate the posterior probability of the data according to the minimum residual sum of squares (RSS), so that 3WN-INB can be used for both discrete and continuous data. After simulation experiments under 10 datasets, 3WN-INB has greatly improved the accuracy and recall rate compared with the traditional model, which verifies that 3WN-INB has better classification performance. In our future work, we will consider the assumption of conditional independence of attributes and consider the use of semi-naive Bayes methods or Bayes network methods to make the conditional independence of each attribute stronger and further enhance the classification performance of the model.

The advantage of this paper is that the new classification utilizes three-way decision and incremental learning, which makes the classifier perform well on different types of datasets and provides a new method for the study of the classification field. Objectively, the limitation of this paper is that the assumption of conditional independence of the naive Bayesian classifier attributes has not been improved, resulting in a slight degradation in classification performance when processing datasets with a large number of attributes.

In the future, we will consider the assumption of conditional independence of attributes and the use of semi-naive Bayesian methods or Bayesian network methods, such as building three-way decision semi-naive incremental Bayesian classifiers and three-way decision Bayesian network classifiers, to further improve the impact between attributes, make the conditional independence of attributes stronger, improve the existing limitations of 3WN-INB and further enhance the classification performance of models.

**Author Contributions:** Conceptualization, Z.Y., J.R., C.Z. and L.W.; data curation, Z.Z. and Y.S.; funding acquisition, L.W.; investigation, Y.S.; methodology, Z.Y. and L.W.; project administration, L.W.; software, Z.Y., J.R. and M.W.; validation, Z.Y.; visualization, Z.Z., Y.S. and M.W.; writing—original draft, Z.Y., J.R. and C.Z.; writing—review and editing, C.Z. and L.W. All authors have read and agreed to the published version of the manuscript.

**Funding:** This research was funded by the Basic Scientific Research Business Expenses of Hebei Provincial Universities (JST2022001), Tangshan Science and Technology Project (22130225G), and Innovation and Entrepreneurship Training Project for College Students in Hebei Province (S202210081055).

**Institutional Review Board Statement:** Not applicable.

**Informed Consent Statement:** Not applicable.

**Data Availability Statement:** Not applicable.

**Conflicts of Interest:** The authors declare no conflict of interest.

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
