# Peer review of "A New Three-Way Incremental Naive Bayes Classifier"

_electronics, doi:10.3390/electronics12071730_

Round 1

Reviewer 1 Report

Dear Authors,

I have attached the review report. Please examine it.

Best regards,

Author Response

1- The authors should have used a grammar check editor for some grammatical errors and misprints.

Answer: Thanks for your suggestion, I have checked the grammar with the grammar checker and it has been corrected.

2- Some datasets whose class number is more than 3 should be simulated.

Answer: Thank you very much, I should really choose some data sets with more than 3 categories. I have re-added the dataset with more than 3 categories for both discrete and continuous data, and re-run the experiment. Experimental results show that our model also achieves better results for multi-category datasets.

3- The authors provided the accuracy and recall metrics for the binary classification. They should utilize the precision and F1-score metrics. Moreover, all metrics should be provided for the multi-class classification.

Answer: Thank you very much for your suggestion. Using precision and F1 score as evaluation indicators is indeed better than using accuracy and recall. I have adopted your opinion and replaced the evaluation indicators with precision and F1 score, which will make the evaluation results more convincing. I also use these indicators to evaluate multiple classifications.

4- They should compare the proposed classifier with the well-known classifiers such as kNN, SVM, Random Forest. Moreover, they should mention the following state-of-the-arts articles related to Machine Learning:

Answer: Thank you for such thoughtful advice. I did not consider the comparison with well-known traditional classification models, this is an oversight on my part. I have added RF, SVM, KNN, MLP and other models to the comparison experiment, which can make the experiment more complete. Also, I have read several of the recent studies you mentioned, and I have already mentioned these excellent studies in papers.

5- Conclusion section should be highlighted by mentioning advantages, disadvantages, and limitations of the study. The future works should be extended also.

Answer: Based on your suggestion, I re-added the advantages and limitations in the conclusion section, and further clarified the direction of future work. Thank you for reading our article.

Reviewer 2 Report

The authors of this paper investigate the effectiveness of a new three-way incremental Naive Bayes classifier in handling uncertain data. They combine three-way decision ideas with the traditional Naive Bayes classifier and use an incremental learning method to solve the problem of data dynamics.

Their findings provide important insights into the classification ability of uncertain data and the optimization of training data in the incremental learning stage. However, their study also raises some questions and concerns that warrant further discussion, such as the lack of direct comparison to existing classifiers in terms of accuracy and speed.

Therefore, I hope to know how fast it is compared to other datasets directly; the experiment could be conducted on a larger dataset.

Thank you for giving me this opportunity to review this paper

Author Response

  • comparison to existing classifiers in terms of accuracy and speed

Answer: Thank you for your valuable comments. According to your suggestion, I added comparative experiments of traditional classifiers such as RF, SVM, KNN and MLP in the comparative experiment section to show the comparison with traditional classifiers. In addition, I also replaced the evaluation indicators with precision and F1-score, which can better reflect the results of model classification. In addition, in section 4.4.3, I added a summary of "algorithm time-consuming analysis" to directly show the running speed of different algorithms on different data sets.

  • the experiment could be conducted on a larger dataset

Answer: Yes, your suggestion is very useful. One of the advantages of Naive Bayesian is that it can deal with small data sets and achieve better results. In the face of large data sets, our classifier should theoretically have more advantages. To this end, we added a large data set for experimentation, which contains 19,020 samples, far exceeding the number of samples in the original experimental data. The results also proved that our model performed very well on large data sets.

Round 2

Reviewer 1 Report

The authors did not make all the technical corrections they claimed in the response letter. They only made grammatical corrections. Under these circumstances, it is not possible to publish the article in the journal as it is.

Author Response

Dear Reviewer,
I am so sorry. In fact, the revised content of my manuscript is consistent with my reply letter, but due to my negligence, I did not mark the revised content or the newly added content in yellow or add annotations, leading you to think that the manuscript only modified the grammar mistake. This is my fault.
I re-uploaded the manuscript with the markup, and this time you can clearly see the changes and additions I have made since I have marked them in yellow.
I apologize again for the inconvenience.
Also, thank you for your review.

Round 3

Reviewer 1 Report

Dear Authors,

In the revised manuscript, all the suggestions have been adopted and the major corrections have been made. Therefore, my evaluation is that the revised version of this manuscript can be published in this journal.

Best regards,